# Enriching Large-Scale Eventuality Knowledge Graph with Entailment Relations

**Changlong Yu**[1]                                   cyuaq@cse.ust.hk
**Hongming Zhang**[1]                              hzhangal@cse.ust.hk
**Yangqiu Song**[1,2]                                  yqsong@cse.ust.hk
**Wilfred Ng**[1]                                        wilfred@cse.ust.hk
**Lifeng Shang**[3]                          shang.lifeng@huawei.com

[1] *CSE, HKUST, Hong Kong, China*
[2] *Peng Cheng Laboratory, Shenzhen, China*
[3] *Huawei Noah's Ark Lab, China*

## Abstract

Computational and cognitive studies suggest that the abstraction of eventualities (activities, states, and events) is crucial for humans to understand daily eventualities. In this paper, we propose a scalable approach to model the entailment relations between eventualities ("eat an apple" entails "eat fruit"). As a result, we construct a large-scale eventuality entailment graph (EEG), which has 10 million eventuality nodes and 103 million entailment edges. Detailed experiments and analysis demonstrate the effectiveness of the proposed approach and quality of the resulting knowledge graph. Our datasets and code are available at https://github.com/HKUST-KnowComp/ASER-EEG.

## 1. Introduction

Large-scale real-world knowledge graph construction is critical to the understanding of human language. Knowledge about noun phrases such as concepts (e.g., "apple" `is-a` "fruit") and named entities (e.g., "France" `BelongsTo` "Europe") have been well captured and represented in modern knowledge graphs such as Freebase [Bollacker et al., 2008], YAGO [Suchanek et al., 2007], and DBpedia [Auer et al., 2007]. On the other hand, how to capture and represent the large-scale knowledge about eventualities (activities, states, and events), which describes how entities and things act, has not been widely investigated. Recently, ASER [Zhang et al., 2020b] proposes to build an eventuality knowledge graph extracted from the raw corpus with carefully designed linguistic patterns and the bootstrapping framework. In ASER, all relations among eventualities are discourse relations such as *Causes* or *Conjunction* and highly confident connectives from Penn Discourse Treebank (PDTB) [Prasad et al., 2008] are used to extract those relations. One limitation of this approach is the missing of some important relations due to the lack of corresponding linguistic patterns. One important relation missing is the entailment relation among eventualities, which describes whether one eventuality $h$ has more general meaning or is inferred by another one $p$ ($p \vDash h$). As suggested by [Zacks and Tversky, 2001], such knowledge reflects how humans abstract the eventualities and could be crucial for a series of eventuality understanding tasks (e.g., future event prediction). Thus in this work, we focus on exploring an efficient way to automatically acquire large-scale eventuality entailment relations.

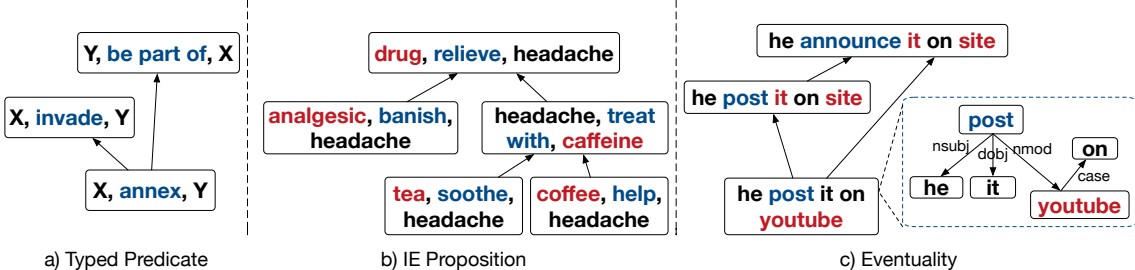

Figure 1: Examples for three entailment graphs with different node types. (a) Typed predicate: is the format of triple $(x, p, y)$ and the predicate $p$ is connected with two argument types, which are denoted with marks (i.e., X and Y) rather than real words. (b) IE Proposition: the instantiated version of typed predicate. (c) Eventuality: a complete semantic graph with multiple dependency relations.

Conventionally, acquiring entailment relations among textual units is known as the entailment graph (EG) construction task, where different works have different definitions of the node (i.e., verb phrase). Examples are shown in Figure 1. [Berant et al., 2011] and [Javad Hosseini et al., 2018] use typed predicates, whose arguments are grounded entity types from KGs such as Freebase, and [Levy et al., 2014] use open-IE propositions, which are binary relations instantiated with two arguments, as the nodes. Different from their simplified definition, the structure of eventuality defined in ASER is more complex. Compared with the typed predicates, an eventuality contains real arguments, which can preserve more specific semantic meanings. Compared with the IE proposition, the eventuality extends to N-ary relations besides binary propositions, which includes more syntactic roles and also more accurate and complete semantic meanings. For example in Figure 1 (c), the node *"he post it on YouTube"* is a small dependency graph itself and contains a prepositional phrase to convey more precise meaning than the proposition *he post it*. At the same time, the complex structure of eventualities also brings extra challenge, which makes the approaches designed for typed predicates and IE propositions not applicable. Besides the node definition, another difference between eventuality and existing EG approaches is the scope. Existing EG construction works often focus on a specific domain such as healthcare [Levy et al., 2014] or email complaints [Kotlerman et al., 2015], where the high-quality expert annotation is available as the training signal. As a comparison, eventualities in ASER are from the open domain, whose knowledge belongs to the commonsense. Compared with the domain-specific knowledge, it is harder to get sufficient human annotation as the training signal for these commonsense knowledge, which could be transferred from linguistic patterns in ASER [Zhang et al., 2020a]. Last but not least, the efficiency of existing EG construction approaches also limits their usage for large-scale KG construction. For example, a standard method used by existing approaches is the Integer Linear Programming (ILP [Berant et al., 2011]), which could only handle 50-100 nodes. It is infeasible to directly apply existing construction approaches to ASER with millions of nodes. Though Javad Hosseini et al. [2018] uses global soft constraints for large-scale graphs, it only generalizes to type predi-

| Node Type | Paper | #Graphs | #Nodes | #Edges | Domain |
|---|---|---|---|---|---|
| Typed Predicate | Berant et al. [2011] | 2,303 | 10,672 | 263,756 | Place/disease |
| | Javad Hosseini et al. [2018] | 363 | 101K | 66M | News |
| IE Proposition | Levy et al. [2014] | 30 | 5,714 | 1.5M | Healthcare |
| Textual Fragment | Kotlerman et al. [2015] | 457 | 756 | 7862 | Email |
| **Eventuality** | Ours | 473 | 10M | 103M | Commonsense |

Table 1: Comparison with existing entailment graphs in terms of the scale and domain.

cates rather than complicated eventualities. The detailed comparisons in terms of scale and domain are shown in Table 1.

In this paper, to address the limitations of existing approaches, we propose a three-step eventuality entailment graph construction method. In the first step, we decompose each eventuality into ($predicate$, the set of $arguments$) pair and the set of $arguments$ could be ($subject, object$), ($subject, object, prepositional\ phrase$) and ($subject, adjective$). And then, in the second step, local inference, which leverages the compositional inference on both the predicates and arguments, is conducted to build up local pair-wise entailment relations. Last but not least, to populate entailment edges globally, we carefully select the predicate entailment paths rather than all the transitive closure and then generalize to eventuality entailment paths. The global entailment inference is conducted along eventuality paths based on local entailment scores in the second step. Using those graph construction techniques, we obtain large and high-quality entailment graphs over eventualities containing more than 103 million edges. To the best of our knowledge, this is the first resource of enormous entailment relations in the general commonsense domain. Our proposed decomposition methods allow for large scale graphs to improve the coverage.

The rest of the paper is organized as follows. In Section 2, we introduce the eventuality entailment graph construction task as well as the notations. In Section 3, we introduce the details about the proposed approach. Implementation details and extensive evaluations are presented in Section 4 and 5 respectively. In the end, we use Section 6 to introduce the related works and Section 7 to conclude this paper.

## 2. Problem Definition

In this section, we formally define the eventuality entailment graph construction task. An eventuality entailment graph (EEG) is a directed graph where nodes are an eventuality $E_i \in \mathcal{E}$ and the edge between two eventualities $E_i$ and $E_j$ represents that $E_i$ entails $E_j$. Following the definition in [Zhang et al., 2020b], each eventuality $E$ is extracted from certain syntactic patterns to keep complete semantics about an activity, state, or event and hence a verb-centric dependency graph. The task is to recognize large-scale high-quality entailment relations among eventualities.

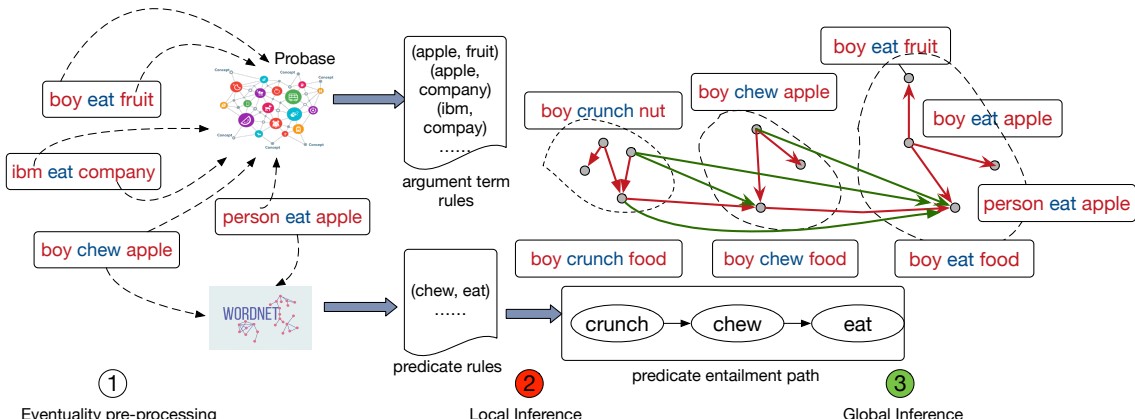

Figure 2: Three-step EEG construction framework. In the first step, eventualities are decomposed into argument sets and predicates, which are mapped to Probase and WordNet to generate inference rules. In the second step, the local inference is conducted to generate entailment relations, which are denoted with red arrows. In the third step, we conduct global inference to further expand the entailment graphs. We indicate the edges from the global inference with green colors.

## 3. Method

To acquire large-scale eventuality entailment knowledge accurately, we proposed a three-step inference framework, whose demonstration is shown in Figure 2. In the first step, we decompose complicated eventuality graphs into argument sets and predicates. Second, we map them to Probase [Wu et al., 2012, Song et al., 2011] and WordNet [Miller, 1998] and conduct local inference to acquire high-quality entailment relations. Third, to acquire more knowledge, we leverage the transitive rules to conduct global inference along the predicate entailment path. Details are introduced as follows.

### 3.1 Eventuality Pre-processing

For each eventuality $E_i$, as the verb can be connected to multiple arguments, it has a multi-way tree structure rather than a binary tree structure like typed predicates or propositions. Following the observations and methods in [Levy et al., 2014], propositions entail if their aligned lexical components entail. We extend from binary propositions to N-ary eventualities and decompose each eventuality $E_i$ into $(p_i, a_i)$ pairs where the predicate $p_i$ is the root node of dependency graph (i.e., center verb) and $a_i = \{t_l^i, l \in 1, ..., L\}$ is the set of arguments that have directed edge with root $p_i$. Table 2 summarizes the eventuality decomposition for the most frequent eventuality patterns in ASER. The number of argument terms in $a_i$ could be 1, 2 or 3, i.e., $L = 1$, 2 or 3. Take the eventuality "he post it on youtube" in Figure 1 (c) as an example, after the decomposition, we will get $p_i = $ post and $a_i = \{$he, it, youtube$\}$. On top of the decomposed eventuality representation, we can then conduct local and global entailment inferences to construct the eventuality graph.

|  | Pattern | Code | Predicate | Argument Set |
|---|---|---|---|---|
| Activities /Events | $n_1$-nsubj-$v_1$ | s-v | $v_1$ | $\{n_1\}$ |
|  | $n_1$-nsubj-$v_1$-dobj-$n_2$ | s-v-o | $v_1$ | $\{n_1, n_2\}$ |
|  | $n_1$-nsubj-$v_1$-nmod-$n_2$-case-$p_1$ | s-v-p-o | $v_1$-$p_1$ | $\{n_1, n_2\}$ |
|  | $(n_1$-nsubj-$v_1$-dobj-$n_2)$-nmod-$n_3$-case-$p_1$ | s-v-o-p-o | $v_1$ | $\{n_1, n_2, p_1\text{-}n_3\}$ |
| States | $n_1$-nsubj-$v_1$-xcomp-a | s-v-a | $v_1$ | $\{n_1, \text{a}\}$ |
|  | $n_1$-nsubj-$a_1$-cop-be | s-be-a | be-$a_1$ | $\{n_1\}$ |
|  | $(n_1$-nsubj-$a_1$-cop-be)-nmod-$n_2$-case-$p_1$ | s-be-a-p-o | be-$a_1$ | $\{n_1, p_1\text{-}n_2\}$ |

Table 2: Eventuality decomposition for different types. Eventualities are decomposed into the format of (*predicate*, set of *arguments*). 'v' stands for verbs except 'be', 'n' for nouns, 'a' for adjectives, 'p' for prepositions, 's' for subjects, and 'o' for objects.

## 3.2 Local Inference

In this section, we present methods of computing local entailment scores based on aligned lexical components of decomposed eventualities. To be specific, we first compute the entailment score for the argument terms and predicates separately and then merge them with a compositional eventuality inference as the final local prediction.

### 3.2.1 ARGUMENT TERM INFERENCE

We first introduce how to construct entailment inference rules among all argument terms $t \in \mathcal{T}$, where $\mathcal{T}$ is the argument term set[1]. Web-scale concept taxonomy Probase [Wu et al., 2012], where each entry is a triple (concept, instance, frequency) extracted from precise patterns, is adopted to look up hypernymy pairs with higher probability. For each $t \in \mathcal{T}$, we conceptualize it and select top $k$ hypernymy candidates *w.r.t.* the co-occurrence probability. An example of the conceptualization for "apple" is {fruit, company, food, brand, fresh fruit} and the hypernyms hit in the $\mathcal{T}$ forms feasible argument term entailment rules such as (apple ⊨ fruit), (apple ⊨ company), or (apple ⊨ food). We use the co-occurrence probability in the Probase as the argument term entailment score denoted by $L_{ij}^t$ between $t_i$ and $t_j$ ($L_{ij}^t = 1.0$ if $t_i = t_j$). We filter out low score pairs based on threshold $\tau$ and recognize inference rules $TR = \{(t_i, t_j, L_{ij}^t)|t_i, t_j \in \mathcal{T}, L_{ij}^t > \tau\}$ among argument terms.

Based on the derived $TR$, we could compute the entailment score between two aligned argument set. Given argument set $a_i = \{t_l^i, l \in 1, ...L\}$ and $a_j = \{t_l^j, l \in 1, ..., L\}$, we define argument set entailment score to be $L_{ij}^a = P(a_i \vDash a_j)$. The probability that argument set entailment holds is the logical OR operation [2] over all the probability that aligned argument

---

1. Based on the eventuality patterns shown in Table 2, the majority of argument terms are typically noun phrases, prepositional phrases, and adjectives.
2. If we adopt logical AND operator, the way of exact string match and match from Probase, may reject a huge volume of semantically related term pairs *i.e.,* $L_{ij}^t = 0$. We prefer to keep all the potential candidates in the first step.

term entailment holds, *i.e.,*

$$P(a_i \vDash a_j) = 1 - \prod^{L}(1 - P(t_l^i \vDash t_l^j)) = 1 - \prod^{L}(1 - L_{t_l^i t_l^j}^t). \tag{1}$$

### 3.2.2 Predicate Inference

We then introduce how to construct entailment relations among predicates $p \in \mathcal{P}$, where $\mathcal{P}$ is the predicate set. One thing worth mentioning is that beyond the single verbs, for some eventuality patterns, we also consider the combination of verb and the associated preposition as the predicate, where the verb itself might be semantically incomplete. One example is the "v-p" combination in pattern *s-v-p-o* such as "take over" from "he takes over the company". After removing predicates whose frequency are less than five, 5,997 single verbs and 13,469 verb-prepositions are left to generate the predicate inference rules. Following [Berant et al., 2011, Tandon et al., 2015], for each predicate $p_i$ except light verbs[3], we extract the verb entailment and direct hypernymy of $p_i$ from WordNet to form the predicate entailment rules such as (know $\vDash$ remember). Considering that verb hierarchy in the WordNet contains non-negligible noises and predicate ambiguity is also common in the extracted raw rules, we define $L_{ij}^p$ to be the entailment score between $p_i$ and $p_j$ to quantify the reliability of the extracted rule $p_i \vDash p_j$. We adopt the asymmetric similarity measure - Balanced Inclusion (BInc) [Szpektor and Dagan, 2008] over the feature vectors of $p_i$ and $p_j$. Since predicate entailment depends on context, *i.e.,* predicates with different argument may have different meaning, we first select the eventualities $E' \in \mathcal{E}$ that share the same arguments between $p_i$ and $p_j$. We gradually augment the context of $p_i$ by choosing the eventuality $E_k = (p_k, a_k)$ with the same predicate $p_i$ whose arguments have larger probability of entailed by $a_i$ than threshold $\lambda$, *i.e.,* $\{E_k \in \mathcal{E} \mid p_k = p_i, L_{ik}^a = P(a_i \vDash a_k) > \lambda\}$. We use point-wise mutual information (PMI) between augmented arguments and the predicate $p_i$ as the feature vector of $p_i$ denoted as $\mathbf{P_i}$. The entailment score $L_{ij}^p$ under $p_i \vDash p_j$ is $BInc(\mathbf{P_i}, \mathbf{P_j})$ and the predicate entailment rules are $PR = \{(p_i, p_j, L_{ij}^p) \mid p_i, p_j \in \mathcal{P}\}$.

### 3.2.3 Compositional Eventuality Inference

After obtaining the inference rules for both the arguments and predicates, the eventuality entailment decisions could be made by their composition. We define $L_{ij}^e$ as the local entailment score between the eventuality $E_i = (p_i, a_i)$ and $E_j = (p_j, a_j)$, and adopt a geometric mean formulation of each aligned components' scores following [Lin and Pantel, 2001]:

$$L_{ij}^e = \sqrt{L_{ij}^p \cdot f_{ij} \cdot L_{ij}^a}. \tag{2}$$

Although predicate entailment scores $L_{ij}^p$ are computed based on similar context (approximately similar types of arguments), the seed eventuality pairs $E'$ from distant supervision may be wrong. For example we start from predicate inference rule (see $\vDash$ think) and extract all the eventuality pairs whose argument pairs are identical such as $E_l =$ *she see towel* and $E_r =$*she think towel*. Here $L_{lr}^e = L_{lr}^p$ but obviously the probability of $E_l \vDash E_r$

---

3. Light verb is a verb that has little semantic content of its own and forms a predicate with some additional expression such as *do, give, have, make, take.*

---

**Algorithm 1:** Global inference algorithm over selected entailment paths.

**Input:** Predicate entailment path $e = (p_1, p_2, ...p_l)$, Argument Term rules $TR$,
        Predicate rules $PR$, threshold $\tau_a$ ,threshold $\tau_e$

```
/* Generalize from predicate entailment paths to eventuality
   entailment paths.                                              */
```

$edge\_set = \varnothing$

**for** $i$ $in$ {1,2, ... l-1} **do**

    Generate bipartite graph $G(p_i, p_{i+1}) = (U_{p_i}, V_{p_{i+1}}, \varepsilon)$

    **for** $c = (e_l, e_r) \in \varepsilon$ **do**

        $e_l = (p_i, a_l)$ $e_r = (p_{i+1}, a_r)$

        **if** $a_l == a_r$ or $(L_{lr}^a > \tau_a$ and $L_{e_l e_r}^e > \tau_e)$ **then**

           $edge\_set = edge\_set \cup \{c\}$

        **end**

    **end**

**end**

Traverse over $edge\_set$ to generate eventuality entailment paths.

---

should be lower. At the same time, we observe that the extracted frequencies of two eventualities in ASER are 26 and 4 respectively. To remedy this issue and based on the above observations, we propose a penalty term $f_{ij}$ to discard the effect of polysemous predicates or inherent wrong extractions of eventuality:

$$f_{ij} = \frac{P(a_i|p_i)}{P(a_j|p_j)}, \tag{3}$$

where the probability of the argument $a_i$ co-occurred with predicate $p_i$ are calculated from the extracted frequency.

### 3.3 Global Inference

In this section, we introduce how to conduct global inference to acquire more edges and make the eventuality entailment graph denser. The backbone of the global inference is the transitive property of entailment relations. For example, if we are aware of $E_i \vDash E_j$ and $E_j \vDash E_k$, it is highly likely that $E_i \vDash E_k$. As the centers of eventualities are verbs, we first construct the entailment chain by predicates and then leverage argument inferences to further enrich it.

    We first decompose the whole graph into predicate-centric sub-graphs and obtain 15,302 predicate inference rules $PR$ from about 19K predicates. The organization of those $PR$ follows the forest-like structure (437 trees totally) and most of the trees have low height (average 3) and 'fat' child nodes. After that, we traverse the trees from the root nodes and get the transitive paths[4] such as (perceive, smell, sniff), (perceive, see, glimpse), and (perceive, listen, hark). And then, we leverage predicate entailment paths $\mathcal{S} = \{(p_1, p_2, ..., p_l)| \ p_i \vDash p_{i+1}, p_i \in \mathcal{P}, i \in \{1, 2, ..., l-1\}\}$ to conduct the transitive inference of eventualities.

---

4. Edges connected to certain root nodes are removed from paths since the root nodes are either light verbs or general verbs like "change", "act", or "move".

For each edge $(p_i, p_{i+1})$ in the entailment path $s \in \mathcal{S}$, we automatically construct a bipartite graph $G(p_i, p_{i+1}) = (U_{p_i}, V_{p_{i+1}}, \varepsilon)$, which fulfills two requirements: (1) All eventualities in $U_{p_i}$ and $V_{p_{i+1}}$ have same or entailed arguments; (2) the predicates of eventualities in $U_{p_i}$ and $V_{p_{i+1}}$ are $p_i$ and $p_{i+1}$ respectively. We use the edge weights as the local entailment eventuality score. One example is shown in Figure 2, $(chew, eat)$ is an edge in the predicate entailment path of $(crunch \vDash chew \vDash eat)$. Hence $U_{chew} = \{$boy chew apple, boy chew food$\}$ and $V_{eat} = \{$boy eat apple, boy eat food$\}$. After we traverse all the edges in the path $s$, we could generalize from the predicate entailment paths to eventuality entailment paths, *i.e.*, (boy crunch food $\vDash$ boy chew food $\vDash$ boy eat food). We filter out eventuality pairs whose entailment scores are less than threshold $\tau_e$ to get rid of the low-quality edges.On top of the collected eventuality entailment paths, we could further expand each node by selecting local inference eventuality relations from argument term rules e.g., (boy crunch nut $\vDash$ boy crunch food) and (boy chew apple $\vDash$ boy chew food) in Figure 2.

The overall algorithm is shown in Algorithm 1, where we start from predicate entailment paths and then generalize to eventuality entailment relations. The time complexity of algorithm is $|\mathcal{S}| \cdot O(n^2)$, where $|\mathcal{S}|$ means the number of predicate entailment paths and $n$ means the summation of all eventualities whose predicate are in the predicate path.

## 4. Implementation Details

We use ASER core version to construct the EEG and only keep the eventualities whose patterns appear in Table 2, which leads to the scale of around 10 millions. After the first step of eventuality pre-processing, we obtain 413,503 argument terms( $|\mathcal{T}| = 413,503$) and 19,466 predicates ( $|\mathcal{P}| = 19,466$), which includes 5,997 single-word verbs. For the second step, we harvest 277,667 argument entailment rules ( $|TR| = 277,667$) from Probase and 15,302 predicate entailment rules ($|PR| = 15,032$) from WordNet. When applying the above entailment rules for compositions, we consider ten possible types of eventuality entailment listed in Table 3. For eventuality pairs with different size of argument set like (s-v-o-p-o) $\vDash$ (s-v-o), we assume (p-o) terms have little effect of entailment hence ignore them. For global inference, we traverse totally 473 predicate entailment trees and get 7,321 predicate entailment paths.

## 5. Evaluation and Analysis

In this section, we present detailed evaluation and analysis to show the effectiveness of the proposed approaches and the value of the resulted resource.

### 5.1 Evaluation Details

Following previous works [Tandon et al., 2015, Zhang et al., 2020b], we employ the Amazon Mechanical Turk[5] to annotate the quality of the collect eventuality entailment knowledge. For each type of entailment rules, we randomly sample 100 eventuality pairs and for each of them, we invite five workers to label whether the former one has more specific meaning than the latter or the latter could be inferred from the former and answer binary yes/no

---

5. https://www.mturk.com

| | # Eventuality | # ER(global) | # ER(local) | Acc (local) | Acc (all) |
|---|---|---|---|---|---|
| s-v ⊨ s-v | 3.3M | 32.7M | 10.7M | 89.1% | 85.7% |
| s-v-o ⊨ s-v-o | 5.3M | 45.2M | 14.8M | 90.1% | 89.3% |
| s-v-p-o ⊨ s-v-p-o | 1.9M | 12.6M | 5.3M | 88.3% | 87.4% |
| s-v-o-p-o ⊨ s-v-o | 0.5M | 0.8M | 0.8M | 91.4% | 90.0% |
| s-v-p-o ⊨ s-v-o | 1.1M | 2.7M | 0.9M | 88.5% | 87.2% |
| s-v-o ⊨ s-v-p-o | 0.9M | 5.4M | 2.2M | 87.8% | 86.7% |
| s-v-o-p-o ⊨ s-v-o-p-o | 2.4M | 3.2M | 2.1M | 89.4% | 88.4% |
| s-v-a ⊨ s-be-a | 0.2M | 0.1M | 0.1M | 97.9% | 97.9% |
| s-be-a-p-o ⊨ s-be-a | 0.8M | 0.4M | 0.4M | 96.0% | 95.8% |
| s-be-a-p-o ⊨ s-be-a-p-o | 0.1M | 0.1M | 0.1M | 95.1% | 94.7% |
| Overall | 10.0M* | 103.2M | 37.4M | 91.4% | 90.3% |

Table 3: Statistics and performance of entailment relations among difference eventuality types. # Eventuality means the total number of unique eventualities. # ER (local) and # ER (global) mean the number of eventuality entailment rules before and after global inference. Acc (local) and Acc (all) are the annotation accuracy before and after global inference.

questions. We consider it to be agreed annotation if at least four annotators out of five select the same answer. As a result, the overall agreement is 93.5%, which indicates that the survey is clearly designed and all workers can clearly understand the task.

## 5.2 Result Analysis

The overall results are shown in Table 3, from which we can make the following observations:

- In terms of different patterns, s-v ⊨ s-v gets the worst performance and the reason might be that the eventualities with unary relations between predicates and arguments contain a certain amount of ambiguity, which conveys incomplete semantics such as (people find, people get). Entailment can benefit from N-ary relations and hence s-v-o-p-o ⊨ s-v-o-p-o achieves much better results.

- For the eventuality inferences of *states*, the perfect accuracy may result from the simple modality of s-v patterns, *i.e.,* linking verbs could naturally entail be-verb such as smell, taste, prove *etc.*

- Global inference has increased the scale of entailment rules by averagely three times while it doesn't lead to the heavy drop of the performance, which proves the effectiveness and usefulness of our proposed global inference algorithm.

## 5.3 Discussion

**Comparison with existing EG construction approaches.** Compared with conventional learning based EG construction methods [Berant et al., 2011, Javad Hosseini et al., 2018], the proposed method is unsupervised, which makes is suitable for handling large-scale

open domain eventualities. Moreover, traditional methods are originally designed for typed predicates and struggle at handling the complex N-ary patterns of eventualities. In terms of the efficiency, especially for the global inference, our algorithm also outperforms previous methods [Javad Hosseini et al., 2018, Berant et al., 2011] due to its simplicity.

**Devised entailment graph/rules.** We have introduced the detailed differences between eventuality and other node types of entailment graphs in Figure 1 as well as Table 1. In our constructed EEG, the devised entailment relations between eventualities would greatly help the reasoning over large-scale ASER and make the original eventuality knowledge graph denser with explicit inference rules. For example the entailment relations could be highly related to other relations *e.g., Cause Reason*, which joint reasoning among different relations might be conducted. Compared with the number of nodes (10M) in ASER and the one of new added entailment edges (103M), enriched ASER is still a little sparser. In the future work, we are going to use devised precise rules as supervision signals to further complete ASER in the way of data enhancement or joint reasoning.

**Error analysis.** We analyze all the false positive examples. Most of errors are due to unclear semantic representations such as the eventuality whose arguments are pronouns, where the ability of disambiguation provided by pronouns is pale. An example is *it leave in water* and *it die in water*. The remaining errors mainly come from the data sparsity, which leads to failure of penalty term imposed on the local eventuality entailment score.

**Potential applications.** As one of the core semantic knowledge, the entailment knowledge among eventualities can be used to help many downstream tasks such as textual entailment and question answering [Javad Hosseini et al., 2018]. From another angle, the acquired entailment knowledge could also serve as the ideal testbed of probing tasks such as general taxonomic reasoning [Richardson and Sabharwal, 2019] and abductive natural language inference [Bhagavatula et al., 2019].

## 6. Related Work

**Eventuality Knowledge Graph.** Traditional event-related knowledge organizations are either compiled by domain experts (FrameNet [Baker et al., 1998]) or crowd-sourced by ordinary people (ConceptNet [Speer et al., 2017]). These resources often face the data sparsity issue due to their small scale. To solve this problem, Tandon et al. [2015] built an activity knowledge graph with one million activities mined from movie scripts and other narrative text. After that, Zhang et al. [2020b] leverages the multi-hop selectional preference [Zhang et al., 2019] about verbs and uses verb-centric patterns from dependency grammars to harvest more than 194 million unique eventualities from a collection of various corpus. By doing so, these two approaches acquire much larger scale eventuality knowledge as they can get rid of the laborious human annotation. As these two approaches manually select linguistic patterns to extract relations among eventualities, many important relations such as the entailment relation are still missing and we still need to devote further efforts to construct a complete eventuality knowledge graph.

**Entailment Graph.** Conventionally, textual entailment has been utilized in the pairwise manner [Dagan et al., 2013] while ignoring the relation dependency. To remedy that, the community started to construct entailment graphs which could help high-order/complex reasoning. Different graphs may have different node definition such as words [Miller, 1998],

typed predicates [Berant et al., 2011, Javad Hosseini et al., 2018], and propositions [Levy et al., 2014] for different purposes. Typically, these graphs are high-quality, domain specific, and small scale because they are carefully annotated by domain experts. Different from them, we directly apply an unsupervised framework to construct entailment relations among open domain eventualities, which is effective and efficient.

## 7. Conclusion

In this paper, we propose a three-step framework of acquiring eventuality entailment knowledge. Compared with existing approaches, the proposed framework can handle much larger scale eventualities without sacrificing quality. As a result, we successfully built entailment relations among a ten-million eventuality knowledge graph. Experiments and analysis prove the quality of the collected knowledge and effectiveness of the proposed framework.

## Acknowledgement

This paper was supported by the Early Career Scheme (ECS, No. 26206717) and the Research Impact Fund (RIF, No. R6020-19) from the Research Grants Council (RGC) in Hong Kong as well as the Gift Fund from Huawei Noah's Ark Lab.

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
