# OpenReview forum: "Enriching Large-Scale Eventuality Knowledge Graph with Entailment Relations"
_AKBC.ws/2020/Conference — AKBC 2020_

### Official Review · AnonReviewer1 · 2020-03-15
**Method for conustrcting high-precision entailment graph**

**Rating:** 6
**Confidence:** 4

**Review:**

This paper proposes a method for automatically constructing large entailment graphs. The method seems reasonable but there are various issues regarding clarity, novelty and empirical evaluation. I think this paper is below the bar of ACL/EMNLP but focuses on a topic that is central to a conference like AKBC and thus is a good fit.

The method proposed has 3 parts:
a) Defining the nodes of the graph which correspond to "events". Unlike prior work, these nodes do not have any variables in them and can be both unary binary and ternary. Authors claim that this is an advantage since the semantic meaning is more precise but there is a big price also - using the rules is harder when they are more specific. Unfortunatley there is no real evaluation that tests whether this is indeed an issue
b) Defining a local similarity score between pairs of nodes. This is based on combining a similarity score between arguments and a similarity score between predicates. The scores are relatively straightforward

One thing I found weird was that the score for a set of arguments was defined using *or* rather than *and*, that is, it seems enough if you have two sets of aligned terms, that only one of them has high similarity to give a high score to the entire set. This seems counterintuitive and is not explained.

c) Defining a procedure for going through entailment paths and adding edges between events. This is done in a fairly precision-oriented manner, as taking the transitive closure can be too noisy. Overall, during the experimental evaluation it is unclear how many edges one gains by adding the global step since in many cases the numbers given are coarse and it seems like very few new rules were added

Overall, the paper proposes a method for constructing graphs and reports some accuracy of generated rules and it seems to be doing ok. I am unsure if this is enough for AKBC, for a first tier conference there are other things that must be done:

1. Some discussion on the usefulness of representation + empirical experiment. The authors change the representation such that it is more specific but this makes it applicability low. It is hard to say whether 100M rules is a lot or not, it could very well be that even though 100M sounds like a lot, trying to use these rules in an application will fail because the rules are too specific. As it is, it is hard to judge the recall of the system, and it is likely that there are many rules it does not capture and it is hard to say how many rules will actually be used in a scenario where one would want to use them.

2. Clarity - various places in the paper were unclear:
3.2.1: the first paragraph is not clear
3.3: Why do the rules create forest? Are there no cyces?

3. Experimental evaluation
Evaluation is post-hoc only, that is you sample rules from the models and estimate precision. But what about recall? What do the rules cover? The authors say this is a large rule-base, but it is hard to judge.

Similarly it seems a better eval. would be to show the rules are useful for some downstream application. Nowadays people are buildling less knowledge-bases of rules since it is an arduous task and moving to encoding knowledge through learning and retrieval on-the-fly. I am not convinced that buildling these KBs would be useful in a NLU task but maybe for testing probes of various sorts.

Also - it seems like in 4/10 cases running the global inference did not add rules or added very little. It is hard to know when the number of rules is reported in millinos only. Why is that?

There is no empirical comparison to prior work as far as I can see.

To conclude:
A method for building entailment graphs is presented resulting in tens of millions of rules that are of high precision. However, the paper does little to convince this is a useful representation and rule-base and empirical evaluation is weak.

---

> ### Author Response · Authors · 2020-04-18
> **Response to Review1**
>
> Thanks for your careful review and constructive suggestions.  We address some of your concerns and questions as follows
>
> Q1: The usefulness of the representation and empirical experiments.
>
> A1:  We have revised the submission to address this question in the Section 5.3  'Discussion - Devised entailment graph/rules' . In short the devised entailment rules could be used to bring richer semantic information for the eventuality knowledgre graph ASER. The enriched ASER would be denser and further help downstream tasks such as dialogue systems, event understanding and narrative generation
>
> Q2:  Clarity of section 3.2.1 and 3.3.
>
> A2:  In terms of Section 3.2.1, we have added the explanations of choosing the logical OR operator instead of the logical AND operator in the revised submission (Please refer to Response to Review2 for one  detailed example).
>
> As for the ‘forest-like structure’ term,  we use it because we are starting from 473 isolated predicate entailment trees, which could be regarded as forest. Even though combining predicates and argument may create new edges among different trees, it is relatively rare. Even though by principle the resulted KG should be a graph rather than a clean forest due to these added edges, to effectively reflect the KG structure property and distinguish it from a normal graph, we call it a graph with ‘forest-like structure’.
>
> Q3: the recall of entailment graph construction and its downstream applications.
>
> A3: It is common that the recall for knowledge graph construction is not well studied or evaluated such as well-known Probase [1], ConceptNet [2]. All the existing KGs are incomplete.
>
> Our strategy is to harvest high-precision entailment pairs as many as possible, which could be further used as supervision signals for KG completion. Hence adding more edges from our constructed EEG could improve the recall. On the other hand, our entailment graph covers most common verbs or verb phrases of English (5,997 single-word verbs and 19,466 verb phrases) , which is much broader than existing resources such as Knowlywood [3] and ConceptNet [2].
>
> ASER, the largest eventuality KG, explicitly models the events and actions in the real world. And our precise method further provides an important relation for the nodes in ASER. The new entailment relation together with discourse relations would help eventuality reasoning even commonsense reasoning to better understand world knowledge. When it comes to usages for downstreaming applications, probing tasks [4] have already used similar resources to test the taxonomic reasoning abilities of existing models. Another usage could be data augmentation, which means that each edge in our KG could be regarded as one sentence with specific knowledge. Augmenting training data or joint reasoning could not be simply replaced by querying from pretrained language models. Despite the powerful representation ability of pretraining,  symbolic rules would still serve as another important inductive bias for the model generalization.
>
> [1] Probase: A Probabilistic Taxonomy for Text Understanding
> [2] ConceptNet 5.5: An Open Multilingual Graph of General Knowledge
> [3] Knowlywood: Mining Activity Knowledge From Hollywood Narratives
> [4] What does my qa model know? devising controlled probes using expert knowledge
>
> Q4:  There are 4 out 10 patterns that seem no more rules added by global inference. Why are the numbers of rules reported as million-level?
>
> A4: The reason why 4 out of 10 patterns in Table 3 have no (obvious) new inferred rules from the global step is that there are no significant transitive properties for those patterns.
>
> For example,  s-v-o-p-o $\vDash$ s-v-o has two forms of global rules (s-v-o-p-o $\vDash$ s-v-o-p-o -> s-v-o) and (s-v-o-p-o $\vDash$ s-v-o $\vDash$ s-v-o). In other words, join-like operators over two patterns make generated rules sparse thus there are few new entailment rules from the perspective of global inference.  (s-be-a-p-o $\vDash$ s-be-a) and (s-v-a $\vDash$ s-be-a) are the similar cases such as (smile $\vDash$ be), (sound $\vDash$ be), (look $\vDash$ be).
>
> The numbers of new added entailment pairs for those four patterns are tens of thousand and thus could not reveal in the millions. Considering the scale of six left patterns, we report the numbers in million in Table 3. We would list the exact numbers of statistics in our released data resource.

---

### Official Review · AnonReviewer3 · 2020-03-27
**This paper improves the prior work.**

**Rating:** 6
**Confidence:** 2

**Review:**

This paper proposes a three-step framework for acquiring eventuality entailment knowledge. Compared with existing approaches, the proposed framework can handle much larger scale eventualities in an unsupervised manner.
However, it would be better to include a case study and compare it with previous work such as ASER.

Comments
- The domain of eventuality is commonsense. Can this paper adopt ConceptNet instead of Probase of WordNet?
- It is better to include a case study and compare it with previous work such as ASER.
- Do hyperparameters (e.g., threshold) affect the result?

This paper improves the prior work, but analysis of the proposed method and comparisons are missing

---

> ### Author Response · Authors · 2020-04-18
> **Response to Review3**
>
> We thank you for the comments and your questions are clarified below.
>
> Q1: Why not adopt conceptNet?
>
> A1:  ConceptNet is a high quality commonsense knowledge resource, but its relatively small scale restricts its usage in our model. For example, ConcepNet 5.0 contains 164 thousand `IsA’ relations while Probase contains 87.6 million `IsA’ relations. Moreover, similar to Probase, ConceptNet also does not have hierarchical relations about verbs and that is why we select WordNet to cover such kind of knowledge.
>
> Q2：Compare with ASER.
>
> A2: As explained in the first paragraph of Section 1, ASER only contains discourse relations between eventualities such as Cause and does not include the important entailment relation. Actually, our work is built on the basis of ASER and enriches the relation type of eventuality knowledge graph. Also new-added accurate edges for ASER in Table 3 would substantially provide richer semantic information.
>
> Q3: hyperparameters affect the result.
>
> A3: As for the hyperparameters, we select it based on the balance trade-off accuracy and coverage. In detail, when conducting the global inference step, we manually construct 200 annotated eventuality pairs as development set to select the hyperparameters (i.e., the threshold for argument inference $\tau_{a}$ and predicate inference $\tau_{e}$) to make sure the induced knowledge is of high quality without sacrificing the coverage too much.

---

### Official Review · AnonReviewer2 · 2020-03-29
**unsupervised method for entailment in eventuality KG**

**Rating:** 7
**Confidence:** 4

**Review:**

This paper focuses on the problem of adding entailment relations in an eventuality (activities, states, events) KG. The main contribution is a pipelined unsupervised method for this problem. The method is divided into three stages: (1) decomposing eventualities -  predicates (mapped to WordNet)  and arguments (mapped to Probase), (2) Local inference step - aggregate entailment scores  on predicates and arguments, and (3) Global inference step - use transitivity of entailments. Human evaluation demonstrates quality of the inferred entailment relations. Overall the paper is well written.

The choice of aggregating scores over aligned arguments (logical OR, Equation 1) is not well motivated. If a single argument matches, the score for the set becomes 1 irrespective of other arguments. Is this expected? Isn’t logical AND more suitable? If not, how does a logical OR produce high quality entailment relations? A discussion on such choices would be really useful.

In Results Analysis (5.2), the lower performance for s-v compared to s-v-o-p-o  is attributed to unary relations between predicates and arguments being ambiguous. However, the difference in performance  is there only for global inference step and not for local inference step. It would be interesting to know why there is more drop in case of s-v.

It would be interesting to know how the method compares on smaller graphs. If the method can’t handle smaller graphs, it would be informative to highlight why and what sizes does it expect.

In the first paragraph of Introduction, it will be useful to define eventuality when it is introduced with the help of an example (E.g., the one in Fig 1).

Fig 2 caption: generated => generate

---

> ### Author Response · Authors · 2020-04-18
> **Response to Review2**
>
> Thank you so much for your valuable comments. We address your questions as follows.
>
> Q1:  Logical OR or Logical AND
>
> A1:  We can use one example to explain why we choose logical OR rather than logical AND.  (A brief explaination has been updated in the revision.)
>
> For example, there are two eventualities: e_i = ‘mary eat apple’; e_j = ‘mary.clark eat fruit’. Hence a_i = {mary, apple},  a_j = {mary.clark, fruit}. Suppose P(mary $\vDash$ mary.clark) = 0 (they are neither matched in Probase nor exactly the same), P(apple $\vDash$ fruit) = 0.89.
>
> We would like to obtain the probability of entailment score P(a_i $\vDash$ a_j) between two argument sets. As used in our approach, if the logical OR operator is adopted, P(a_i $\vDash$ a_j) =0.89. However, if the logical AND operator is adopted, P(a_i $\vDash$ a_j) would become 0.
>
> As we already applied a relatively strict matching rule (i.e., string match), which may reject a huge volume of semantically related but exactly not same argument term pairs, we prefer to keep all the potential candidates in the first step.  After that, we leverage stricter operators in the following steps to control the overall good quality, which is shown in Table 3.
>
> Q2:  s-v performance drop for global inference.
>
> A2:  The main reason for the ‘s-v’ pattern’s performance drop after the global inference is the relatively low quality of global inference compared with the very-high accurate local inference.
>
> As shown in Table 3, the global inference step for pattern ‘s-v’ could acquire entailment pairs three times than the local inference step (from 10.7M to 32.7M). As a comparison, for pattern ‘s-v-o-p-o’,  global inference step only increases half of entailment pairs than the local inference (from 2.1M to 3.2M).
>
> As we randomly sample all the annotated examples, after the global inference, most of the annotated ‘s-v’ examples are coming from the global inference step while only a small portion of the ‘s-v-o-p-o’ examples are from the global inference. This can help explain why we got a larger performance drop for the `s-v’ pattern.
>
> Q3:  Compare the methods that work on small graphs
>
> A3:  Eventuality entailment construction on small graphs is a different research problem because we can annotate a small amount of data and train a good supervised model.However, such a solution is not feasible for large graphs.
>
> As our goal is to create a complete eventuality entailment graph, which must have a very large scale and may require a huge amount of annotations for a good supervised model.That is exactly why we want to propose an unsupervised problem to handle this challenge.

---

### Decision · Program_Chairs · 2020-04-30

**Decision:**

Accept

**Comment:**

This paper proposes a novel framework for acquiring eventuality entailment knowledge to construct a knowledge graph. The multi-step construction process is well explained and has clear justification. However, the paper could be stronger if it expands more on convincing audience that such knowledge graph is a useful representation, has promising downstream applications. The work can also benefit from adding more empirical evaluation.